

# Impact of trace metal concentrations on coccolithophore growth and morphology: laboratory simulations of Cretaceous stress.

Giulia Faucher[1], Linn Hoffmann[2], Lennart T. Bach[3], Cinzia Bottini[1], Elisabetta Erba[1], Ulf Riebesell[3]

[1] Earth Sciences Department "Ardito Desio", Università degli Studi di Milano, Milan, Italy

[2] Department of Botany, University of Otago, Dunedin, New Zealand

[3] Biological Oceanography, GEOMAR Helmholtz Centre for Ocean Research Kiel, Kiel, Germany

*Correspondence to*: Giulia Faucher (giulia.faucher@unimi.it)

**Abstract.** The Cretaceous ocean witnessed intervals of profound perturbations such as volcanic input of large amounts of $CO_2$, anoxia, eutrophication, and introduction of biologically relevant metals. Some of these extreme events were characterized by size reduction and/or morphological changes of a few calcareous nannofossil species. The correspondence between intervals of high trace metal concentrations and coccolith dwarfism suggests a negative effect of these elements on nannoplankton biocalcification process in past oceans. In order to verify this hypothesis, we explored the potential effect of a

mixture of trace metals on growth and morphology of four living coccolithophore species, namely *Emiliania huxleyi, Gephyrocapsa oceanica, Pleurochrysis carterae* and *Coccolithus pelagicus*. These taxa are phylogenetically linked to the Mesozoic species showing dwarfism under excess metal concentrations. The trace metals tested were chosen to simulate the environmental stress identified in the geological record and upon known trace metal interaction with living coccolithophores algae.

Our laboratory experiments demonstrated that elevated trace metal concentrations not only affect coccolithophore algae production but, similarly to the fossil record, coccolith size and/or weight. Smaller coccoliths were detected in *E. huxleyi* and *C. pelagicus,* while coccoliths of *G. oceanica* showed a decrease in size only at the highest trace metal concentrations. *P. carterae* coccolith size was unresponsive for changing trace metal amounts. These differences among species allow to discriminate most- (*P. carterae*), intermediate- (*E. huxleyi*), and least- (*C. pelagicus* and *G. oceanica*) tolerant taxa. The

fossil record and the experimental results converge on a selective response of coccolithophores to metal availability. These species-specific differences must be considered before morphological features of coccoliths are used to reconstruct paleo-chemical conditions.



## 1 Introduction

Trace metal concentrations influence the productivity and species composition of marine algae communities (Bruland et al., 1991; Sunda and Huntsman, 1998). A number of trace metals are fundamental micronutrients (e.g. zinc, iron, copper, nickel) but some of them can become toxic and inhibit marine alga productivity at elevated concentrations. Others like lead and
mercury have no known metabolic functions and can affect marine phytoplankton growth already at low concentrations.

The geological record offers the opportunity to investigate past case histories marked by profound changes in the ocean, such as volcanic injection of large amounts of $CO_2$, ocean anoxia, eutrophication, and introduction of biologically relevant metals (e.g. Larson and Erba, 1999; Erba, 2004; Jenkyns, 2010; Erba et al., 2015). These perturbations can be seen as "natural experiments" useful to decrypt the ecosystem response to major perturbations at time scales longer than current
modifications. Finding out how the changes in seawater composition affect marine biota requires the integration of a long–term and large–scale geological perspective that has been recognized as an essential ingredient for more coherent predictions of how marine organisms might react to future environmental changes. Insights on ocean/atmosphere dynamics under warmer-than-present-day conditions predicted for the end of this century can only be obtained by including geological data of past ecosystems, especially those derived from cases of extreme conditions. Well-known perturbations were the "Oceanic
Anoxic Events" (OAEs) which took place during the Mesozoic. These events were caused by intense volcanism that produced large igneous provinces (LIPs) (Snow et al., 2005; Neal et al., 2008; Pearce et al., 2009; Erba et al., 2015) that released magmatic fluids delivering metals, mixed with warmed ambient seawater that had enough buoyancy to rise to the surface (Snow et al., 2005; Erba et al. 2015). During the latest Cenomanian OAE 2 for example, less volatile elements, such as nickel and iron (released during the formation of the Caribbean LIP), increased by about 8-20 times the background level
while more volatile elements like lead and cadmium (derived from water-rock exchange reactions), increased by about 4-8 times the background levels (Orth et al., 1993; Snow et al. 2005). Entering the ocean environment, more and less volatile elements became biologically and chemically significant as evidenced by changes and turnover in marine plankton community (Leckie, 1985; Leckie et al., 1998; Erba, 2004; Erba et al., 2015).

Studies on calcareous nannofossils documented a size reduction of some coccolithophore species coeval with trace metal
concentration peaks across both the early Aptian OAE 1a and latest Cenomanian OAE 2 (Erba et al., 2015; Faucher et al., 2017). The fossil record shows that, although no Mesozoic nannoplankton taxa survived the mass extinction event at the end of the Cretaceous, reconstructed morphological phylogeny indicates a direct link between selected Mesozoic groups and some living coccolithophores (Bown et al., 2004).

The four species tested, namely, *Emiliania huxleyi, Gephyrocapsa oceanica, Coccolithus pelagicus* and *Pleurochrysis*
*carterae*, evolutionary diverged from one another since the Late Cretaceous, with the exception of *E. huxleyi* and *G. oceanica* that are separated since 250,000 years ago. Specifically, genera *Emiliania* and *Gephyrocapsa* belong to the Cenozoic family Noelaerhabdaceae derived from the extinct Prinsiaceae that, in turn, branched off the Mesozoic family Biscutaceae. Indeed, coccolith dwarfism was observed in genus *Biscutum* during times of high $CO_2$ and metal concentrations





in both OAE 1a and OAE 2 (Erba et al. 2010; Faucher et al 2017). It is challenging to unambiguously disentangle the cause/s of such changes in the fossil record but evidence of a correspondence between intervals of high trace metals concentrations and coccolith dwarfism suggest a negative effect of these elements on nannoplankton biocalcification processes.

Previous work on the response of living coccolithophores to trace metal concentrations focused on *Emiliania huxleyi*. These

studies documented a decreasing growth rate under high trace metal concentrations (Vasconcelos et al., 2001; Hoffmann et al., 2012, Santomauro et al., 2016). So far, no studies have been performed on other coccolithophore species. Furthermore, to our knowledge, this is the first study investigating the effect of high trace metal concentrations on coccolithophore and coccolith morphology and size.

The trace metal tested were chosen based on peaks identified in the geological record (Snow et al., 2005) and known trace

metal interactions with living coccolithophores to simulate the environmental conditions during OAEs.

The main goal of this study is to understand if, similarly to the fossil record, anomalously high quantities of essential and/or toxic metal induce changes in coccolith shape and size and cause coccolith dwarfism in coccolithophore species.

More specifically, we address the following questions: i) does coccolithophore growth change in response to increasing trace metal concentration? ii) does coccolith size and morphology, as well as coccolithophore size, change in response to high and

anomalous trace metal concentrations? iii) do trace metal combinations, which mimic OAEs conditions, lead to a unique response among species or to species-specific responses on morphological features? iiii) do coccolith morphometrical features have some theoretical potential to serve as a proxy to reconstruct trace metal concentrations in sea-water?

## 2 Material and Methods

### 2.1 Culture conditions

Monospecific cultures of the coccolithophores *Emiliania huxleyi, Gephyrocapsa oceanica, Coccolithus pelagicus,* and *Pleurochrysis carterae* were grown as batch cultures in artificial sea water (ASW) produced as described by Kester et al., 1967. The artificial seawater medium was enriched with 64 µmol kg⁻¹ nitrate, 4 µmol kg⁻¹ phosphate to avoid nutrient limitations, f/8 concentrations for vitamins (Guillard and Ryther, 1962), 10 nmol kg⁻¹ of $SeO_2$ and 2 ml kg⁻¹ of natural North Sea water (Bach et al., 2011). The carbonate chemistry was adjusted by bubbling with $CO_2$-enriched air overnight to

reintroduce inorganic carbon and decrease the pH. All culture bottles were manually and carefully rotated three times a day, each time with 20 rotations in order to avoid cell settling. In the control treatment, the medium was enriched with f/8 concentrations for trace metals (Guillard and Ryther, 1962).

Pb, Zn, Ni and V concentrations were added in low (L), medium (M), high (H) and extreme treatments because of their high concentrations identified in the Aptian OAE 1a (Erba et al., 2015) and Cenomanian-Turonian OAE 2 (Snow et al., 2005).

(Table 1.). The trace metal chelator EDTA (ethylenediamine tetraacetic acid) was added to the trace metal stock solutions in order to guarantee a constant level of bioavailable trace metals for phytoplankton and prevent metal precipitation. The





cultures were incubated in a thermo constant Rumed climate chamber (Rubarth Apparate GmbH) at a constant temperature of 15° C, a 16:8 [hour:hour] light/dark cycle, at a photosynthetic active radiation (PAR) of 150 µmol photons $m^{-2} s^{-1}$.

The cultures were pre-acclimated to experimental conditions for 7-10 generations, which varied between 6-10 days depending on the species-specific cell division rates. Monospecific cell cultures of *E. huxleyi* (strain RCC 1216), *G.*

*oceanica* (strain RCC 1303), *C. pelagicus ssp. braarudii* (strain PLY182G) and *P. carterae* were incubated in autoclaved 500 mL square glass bottles (Schott Duran). The initial cell density was relatively low with ~ 50 cells $ml^{-1}$. Final samples were taken when cells were still in their exponential growth phase and cell numbers were low enough to avoid a strong change in the chemical conditions of the growth medium. Therefore, the experimental duration differed among species and among treatments due to the different growth rates. Each treatment was replicated three times. Final cell densities in the

main experiment didn't exceed 50000 cell $ml^{-1}$ in *E. huxleyi*, 20000 cell $ml^{-1}$ in *G. oceanica*, 3000 cell $ml^{-1}$ in *C. pelagicus* and *P. carterae*.

### 2.2 Cell abundance and growth rate

Samples for cell abundance were taken every second day with the exception of the control treatment where samples were

only taken at the end of the experiment. Every sample was gently turned 10 times in order to obtain a homogenous suspension of the cells before sampling. Cell numbers were immediately measured three times without addition of preservatives using a Beckman coulter Multisizer. Specific daily growth rates (µ) were calculated from the least-squares regression of cell counts versus time during exponential growth (Eq.1)

$$\mu = \frac{lnc_1 - lnc_0}{t_1 - t_0} \qquad (1)$$

were $c_0$ and $c_1$ are the cell concentrations at the beginning ($t_0$) and at the end of the incubation period ($t_1$), respectively.

### 2.3 Coccosphere and cell sizes

Cell abundance samples were acidified with 0.1 mmol $L^{-1}$ HCl to dissolve all free and attached coccoliths and subsequently measured 3 times in order to obtain cell diameters and volumes (Müller et al., 2012). The coccosphere volume was estimated

as:

Coccosphere volume = cell volume – coccosphere and cell volume.

Free coccolith volume and coccolith concentrations were also determined for *C. pelagicus* (Fig. 1).




### 2.4 Coccolith dimensions and malformations

#### 2.4.1 Scanning Electron Microscopy (SEM)

Samples were taken for each of the 48 sample bottles. 5-10 ml of sample were filtered by gravity on polycarbonate filters (0.2 µm pore size) and dried directly after filtration at 60°C. Samples were sputtered with gold-palladium. SEM analyses was

performed at the Earth Sciences department of the University of Milan with SEM Cambridge Stereoscope 360. For each sample, 50 specimens were digitally captured and subsequently analyzed with Image J software. All pictures were taken with the same magnification (5000x) and the size bar given on SEM pictures was used to calibrate distances (Fig 2).

#### 2.4.2 Coccolith dimensions and *E. huxleyi* malformation

We measured the length of distal shield (DSL) and the width of the distal shield (DSW) manually using the public domain

program Fiji (Schindelin et al., 2012) a distribution of ImageJ software (Schindelin et al., 2012). The distal shield area (DSA) was calculated, assuming an elliptical shape of the coccolith, as (2):

$$DSA = \Pi \frac{DSL \; x \; DSW}{4} \quad (2)$$

Assuming elliptical shape has been shown to yield near identical results to direct measurements of DSA in *E. huxleyi* (Bach et al., 2012).

*E. huxleyi* malformations were determined by visual comparisons of 100 individual coccoliths for every sample. We sorted the degree of malformation in several categories: our categories were used to describe the morphology of *E. huxleyi* as "normal", "malformed", "incomplete" and "incomplete and malformed" coccoliths (Langer et al., 2010; Langer et al., 2011)

(for reference images for the categories, see Fig 3). We considered normal *E. huxleyi* coccolith with regular shape with well-formed distal shield elements forming a symmetric rim; malformed: malformed coccolith shape or shape of individual elements; incomplete: coccolith with variations in its degree of completion; malformed and incomplete: coccolith with malformed shape and variations in its degree of completion.

### 2.5 Statistics

Prior to statistical analyses, data were tested for normality and homogeneity of variances. To test the null hypothesis the average value of parameters from triplicate cultures were compared between treatments. Mean µ values, coccosphere and cell diameters, coccosphere volume and coccolith sizes of each treatment were compared to the control and among each other. A one-way analysis of variance was used to determine statistical significance of the main effect of trace metals on the variables. A Tukey post-hoc test was used to identify the source of the main effect determined by ANOVA to assess whether

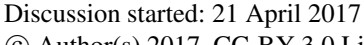
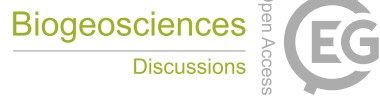


differences in μ and sizes between trace metal treatments were statistically significant. Statistical treatments of data were performed using R software. Statistical significance was accepted for $p < 0.05$.

## 3 Results

### 3.1 Growth rate

In the treatment with extreme trace metal concentrations up to 8 μmol L$^{-1}$, the conditions were so poisonous that all the four species tested didn't survive the acclimation phase. On the other hand, *E. huxleyi*, *G. oceanica*, *C. pelagicus* and *P. carterae* survived in L, M and H. However, the additional introduction of trace metals decreased the growth rate of *E. huxleyi*, *G. oceanica,* and *C. pelagicus* compared to the control treatment. *E. huxleyi* growth rate was 1.22 in the controls and 1.12 in L (Table 2.; Fig 4a). Growth rate in M was 1.16 in all the replicates while H trace metal concentration treatment showed the
lowest growth rate values of 1.10 (Fig 4a).
*G. oceanica* growth rate was 0.66 in the controls. In L, M and H the growth rate was significantly lower compared to the control and respectively of 0.58 in L, 0.60 in M and 0.58 in H. (Table 2.; Fig 4b).
*C. pelagicus* had an average growth rate in the control experiment of 0.56. The growth rate was significantly lower in L, M and H compared to the control. In L, the growth rate is 0.42 while in M and H is 0.43 in average (Table 2.; Fig 4c)
Contrarily, *P. carterae* showed an increase in growth rate with trace metal addition compared to the control. In the control treatment, the growth rate was 0.52 and is significantly lower compared to L, M and H treatments that have a growth rate of respectively 0.57, 0.56 and 0.57 (Table 2.; Fig 4d).

### 3.2 Coccosphere and cell sizes and coccosphere volume

In *E. huxleyi,* the mean coccosphere diameters were significantly lower in the L, M, and H treatments compared to the
control (mean diameter 4.88 μm; Table 2., Fig 5a). The coccosphere volume (= cell volume – coccosphere and cell volume) was reduced under all increased trace metal treatments compared to the control conditions (Table 2.; Fig 6a) with lowest coccosphere volumes recorded in the M treatment.
In *G. oceanica,* the coccosphere diameters were largest in the control treatment (mean diameter 7.25 μm; Table 2.) (Fig 5b). L, M and H coccosphere diameters were significantly smaller compared to the control. Specifically, L and M show similar
values of 6.58 μm and 6.60 μm, respectively, while H shows the lowest coccosphere diameter of 6.14 μm (Table 2.). Similarly, cells diameters were significantly larger in the control treatment (5.45 μm), intermediate in L and M (L= 5.18 μm; M=5.19 μm) and smaller in H (4.74 μm). Furthermore, the coccosphere and cell diameters were significantly smaller in H compared to L and M. The coccosphere volume was significantly reduced under increased trace metal concentrations compared to control conditions (Fig 6b) with similar coccosphere volumes recorded in both L, M and H.



*C. pelagicus* coccosphere and cell diameters were significantly larger in the control (19.82 µm and 15.65 µm, respectively) compared to L (17.12 µm and 10.10 µm, respectively), M (17.05 µm and 10.46 µm, respectively) and H (16.85 µm and 10.38 µm, respectively) (Table 2.; Fig 5c). However, a general increase in coccosphere volume is observed in L, M and H compared to the control treatment. (Fig 6c): the coccosphere volume values are significantly higher in the trace metal treatments compared to the control treatment (Table 2.).

*P. carterae* in the control experiment showed smaller coccosphere diameter in the control compared to L, M and H. The coccosphere size in L and H (coccosphere diameter L= 12.11 µm; H=11.99  µm) is significantly bigger compared to the control (coccosphere diameter 11.70 µm). In M, the coccosphere has a mean diameter of M=11.88  µm (Fig 5d). On the other hand, the cells were very similar among all experiments. The coccosphere volume was lower in the control replicates and similar and only slightly higher in the other three treatments (Table 2.; Fig 6d).

### 3.3 Coccolith size and *C. pelagicus* coccolith concentrations

*E. huxleyi* coccoliths were longer and wider in the control treatment compared to the other three treatments (Table 3.). Increasing trace metal content reduced coccolith length and width significantly ($p < 0.05$) and the high trace metal treatment shows the lowest DSL and DSW coccolith size.

In *G. oceanica* coccolith were longer and wider in the controls compared to the other three treatments (Table 3.). However only in H, coccolith were significantly smaller ($p < 0.05$) compared to the control treatment.

*C. pelagicus* coccoliths in the L, M and H trace metal treatments were significantly smaller compared to the control replicates (Table 3.). A higher number of free coccoliths was present in the trace metal treatments compared to the control replicates (Table 4.). Free coccoliths progressively increased with increasing trace metal content.

*P. carterae* coccoliths showed very similar sizes in all the four experiments (Table 3.)

### 3.4 *Emiliania huxleyi* coccolith malformation

Scanning electron microscope analyses of *E. huxleyi* coccoliths showed changes in the proportion of malformed and incomplete coccoliths. Specifically, malformations and incomplete coccoliths of *E. huxleyi* increased in all trace metal treatments (L, M and H concentrations) by about 20-35 % compared to the control treatment (Fig 7).




## 4 Discussion

### 4.1 Growth rate

Previous studies on *E. huxleyi* responses to trace metal enrichment resulting from volcanic ash, showed no significant effects on growth rate for most ashes tested (Hoffmann et al., 2012). Only the addition of pumice, which released low concentration

of trace metals, had a beneficial effect and increased *E. huxleyi* growth. In one case, however, progressively increased ash content strongly suppressed the growth rate of *E. huxleyi* in the volcanic ash which contained the highest trace metal concentrations (e.g. Pb from 0.5 up to 2.6 nM $L^{-1}$; Ni from 12 up to 60 nM $L^{-1}$) (Hoffmann et al., 2012). Vasconcelos et al. 2001, report a 10-20 % growth rate reduction of *E. huxleyi* with increasing Pb up to 0.25 µmol $L^{-1}$ without additions of EDTA.

In our study, *E. huxleyi* growth, decreased with increasing Pb, Zn, Ni, and V concentrations and the highest concentration of trace metals up to 0.8 µmol $L^{-1}$ slowed down *E. huxleyi* growth by about 10% (Fig 4a).

Trace metal effects on *G. oceanica* have not been tested before but in this study, a decrease of *G. oceanica* growth of about 12% occurred under the highest trace metal concentrations (Fig 4b). A similar toxic response to elevated trace metal concentration was observed for *C. pelagicus* where the growth rate decreased by about 31% in each of the trace metal

treatment compared to the control (Fig 4c). The bigger growth rate reduction observed for *C. pelagicus* suggests a high sensitivity of this species to trace metals enrichment. A stepwise increase in trace metal concentration did not induce any progressive growth rate reduction attesting a strong sensitivity of both *G. oceanica* and *C. pelagicus* already at low trace metal concentrations.

*P. carterae* growth rate generally increased with trace metal concentration (Fig 4d). This beneficial effect of abnormally high

trace metal quantities (L, M and H) on *P. carterae* growth rate might be due to the habitat of this species which are eutrophic lagoons and estuaries (Heimdal, 1993). Since trace metal concentrations are much higher in lagoons and coastal area, trace metal additions would not have expected to have significant effects on this species.

### 4.2 Morphometrical analyses

The coccolithophore species tested evidenced a detrimental effect of trace metals on coccosphere, cell and coccolith sizes.

Indeed, three species, *E. huxleyi*, *G. oceanica* and *C. pelagicus,* evidenced significant size reductions when grown under anomalously high trace metal concentrations. However, the morphometrical responses are highly variable among species (Fig 8): 1) *E. huxleyi*, under high trace metal concentration, reduced its coccolith sizes. This reduction can explain the coccosphere diameter decrease and the concomitant stable cell sizes. 2) Trace metal concentration influenced *G. oceanica* coccosphere and cell sizes. Furthermore, an extra size reduction of both parameters, that goes along with coccolith size

decrease, occurred with the highest trace metal concentration tested. This imply a further noxious effect of very high trace



metal concentration on *G. oceanica* growth. 3) *C. pelagicus* coccosphere, cell diameters and coccolith sizes are negatively influenced by higher trace metal quantities. Increased trace metal induced both coccosphere, cell and coccolith sizes reduction of *C. pelagicus* in all the concentration tested. However, coccolith volume significantly increased under high trace metal concentrations. A plausible explanation is that size decline of the cell goes hand in hand with an increase in the

coccolith numbers that cover the cell. Indeed, progressively increased number of free/detached *C. pelagicus* coccoliths go together with a gradual increase in trace metal concentrations. (Table 4.). This hint a beneficial effect of trace metal on the number of *C. pelagicus* coccolith produced per cell (Paasche et al., 1998; Müller et al., 2012). 4) On the contrary*, P. carterae* doesn't show any sensitivity to trace metal concentration since coccosphere, cell and coccolith sizes remain stable in all the treatment tested. Trace metal concentration in coastal area are commonly much higher compared to the open ocean.

Therefore, due to habitat of this species, which are eutrophic lagoons and estuaries, (Heimdal, 1993), addition of trace metal might not have significant effects for *P. carterae.*

Coccolithophore algae therefore, respond differently to changes in trace metal concentrations. This species-specific sensitivity suggests a different degree of adaptive potential of the species tested.

## 4.3 Analogy and contrast with the fossil record

The trace metal tested, were chosen based on metals peak identified in the Aptian OAE 1a (Erba et al., 2015) and latest Cenomanian OAE 2 (Snow et al., 2005). Zn and Pb are more volatile elements that are concentrated in magmatically degassed fluids. On the other hands, Ni and V are found in higher concentrations in water-rock exchange reactions of typical steady-state hydrothermal vents (Rubin, 1997). Therefore, the mixture of trace metal tested tried to simulate OAE conditions. We emphasize that the coccolithophore species chosen for this experiment are linked to the Mesozoic family Biscutaceae

based of the fossil record tracing their biocalcification history back to ~200 million years ago (Bown et al., 2004). When genomic data-sets are considered for reconstruction of coccolithophore evolution, it appears that the selected Coccolithales order (*C. pelagicus* and *P. carterae*) diverged from the Isochrysidales order (*E. huxleyi* and *G. oceanica*) in the earliest Triassic (De Vargas et al., 2007) or even in the latest Permian (Liu et al., 2010), some 300 million years ago.

It is therefore possible with caution, to try to make a comparison among fossil and living coccolithophore responses since the

species tested in this study have a long common evolutionary history.

Morphometric analyses of selected nannofossil taxa across Cretaceous OAEs in various geological settings, revealed differential species-specific patterns. *Biscutum constans*, a cosmopolitan coccolithophore species of the Cretaceous ocean, evidenced size variations in times of environmental change. Specifically, coccolith dwarfism (*sensu* Erba et al., 2010) occurred at intervals characterized by high trace metal concentrations (Erba et al., 2010; Erba et al., 2016; Faucher et al.,

2017. On the other hand, *Watzanaueria barnesiae*, a cosmopolitan species described as a r-selected opportunistic species (Hardas et al., 2007) remained statistically steady in size across OAEs (Erba et al., 2010; Bornemann and Mutterlose, 2006; Lübke and Mutterlose, 2016, Faucher et al., 2017). Indeed, a more pronounced ellipticity, interpreted as evidence of




malformation, was documented by Erba et al., 2010 during OAE 1a, but not during other times of global anoxia (Bornemann and Mutterlose, 2006; Faucher et al., 2017). The very little variability in *W. barnesiae* size indicates that this taxon was most adaptable and only marginally affected by the paleoenvironmental stress characterizing Cretaceous OAEs. Finally, *D. rotatorius* and *Z. erectus*, species with a meso-eutrophic preference, evidenced inconsistent size trends without proving any

relationship between size and metal peaks (Faucher et al., 2017).

Our laboratory experiments on living coccolithophore algae, demonstrate that elevated trace metal concentrations not only affect coccolith production but, similarly to the fossil record, coccolith size and/or weight. Moreover, as quantified in nannofossil assemblages, our results reveal a species-specific response. In fact, large differences were observed between species and discriminated most- (*P. carterae*), intermediate- (*E. huxleyi, G. oceanica*), and least- (*C. pelagicus*) tolerant taxa

to trace metal content. Parallel changes among fossil and living coccolithophores suggest that trace metal concentrations have the potential to influence coccolith production and sizes.

We stress the fact that both the fossil record and the experimental results converge on a species-specific response of coccolithophores to metal availability. Consequently, the broad use of coccolith sizes as a proxy of trace metal concentration in sea-water should be avoided. Instead, it is crucial to identify the species index/indices that better trace paleoenvironmental

stress induced by excess (selected) metal amounts.

## 5 Conclusion

With this study, we demonstrated for the first time that a mixture of trace metals affected growth and morphology of all of the coccolithophore species tested. A size reduction of the coccosphere and cell diameters has been observed in three of the analyzed species. Furthermore, we evidenced the production of dwarf coccoliths (*sensu* Erba et al., 2010) with high trace

metal quantities. Our data show a species-specific sensitivity of coccolithophore algae to trace metal concentration, allowing the recognition of most- (*P. carterae*), intermediate- (*E. huxleyi* and *G. oceanica*), and least- (*C. pelagicus*) tolerant taxa.

The comparison of living coccolithophore data and Mesozoic calcareous nannofossils shows strong similarities, suggesting that laboratory simulations of past extreme conditions are viable when extant taxa are phylogenetically linked to extinct fossil species. Our study supports the hypothesis that anomalous trace metal conditions in the past oceans significantly

contributed to the morphological coccolith changes during Cretaceous OAEs.

Laboratory experiments on modern coccolithophore species remain the only means to quantitatively assess the individual or combined role of environmental parameters (e.g. trace metal content) on coccolith secretion. The species-specific differences must be considered before coccolith morphological features are used to reconstruct paleo-chemical conditions.



## 6 Acknowledgments

The research was funded through MIUR-PRIN 2011 (Ministero dell'Istruzione, dell'Università e della Ricerca–Progetti di Ricerca di Interesse Nazionale) to E. Erba, and through SIR-2014 (Ministero dell'Istruzione, dell'Università e della Ricerca–Scientific Independence of young Researchers) to C. Bottini. G. Faucher was supported by Fondazione Fratelli Confalonieri.

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



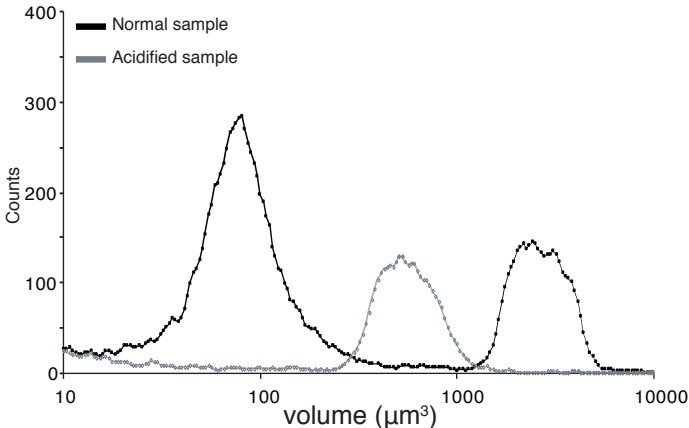

**Figure 1: Example of multisizer volume spectra. Black line: spectra of *C. pelagicus* population (coccolith spectrum and coccosphere spectrum); grey line: spectra of *C. pelagicus* cell after treatment with HCl.**



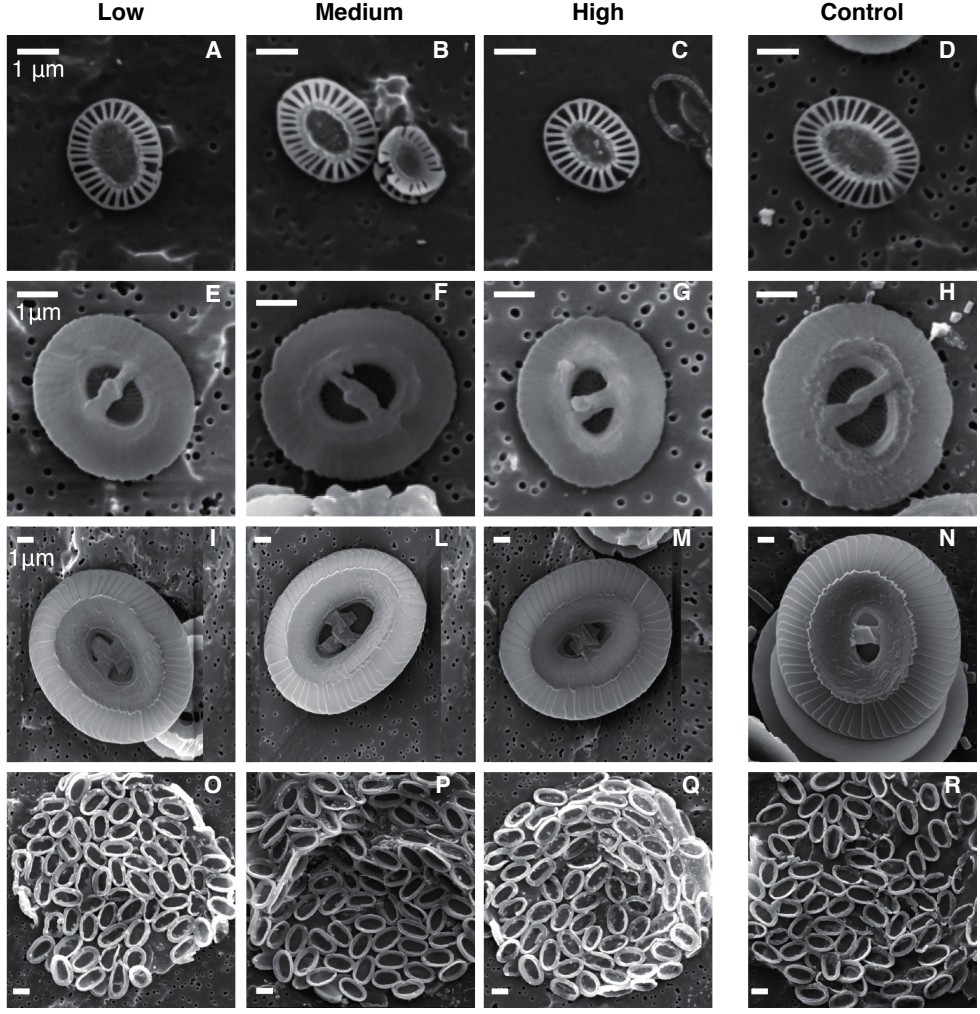

**Figure 2: Plate. Example of the coccoliths of the four species tested under different trace metal concentrations. A-D, *E. huxleyi*; E-H, *G. oceanica*; I-L, *C. pelagicus*; M-P, *P. carterae*.**



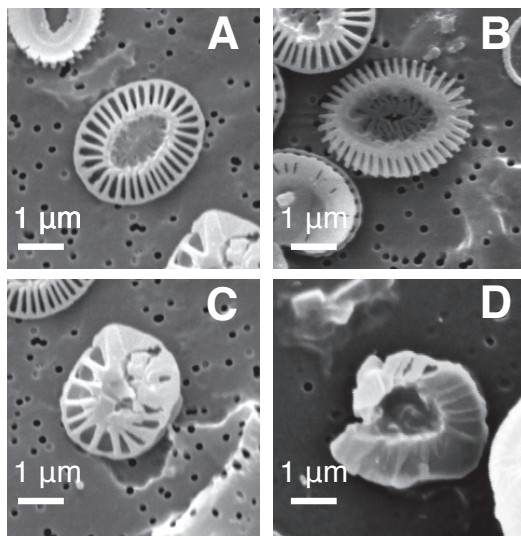

**Figure 3: SEM images of *Emiliania huxleyi* coccoliths. A. Normal coccolith; B. Incomplete coccolith; C. Malformed coccolith; D. Malformed and incomplete coccolith.**




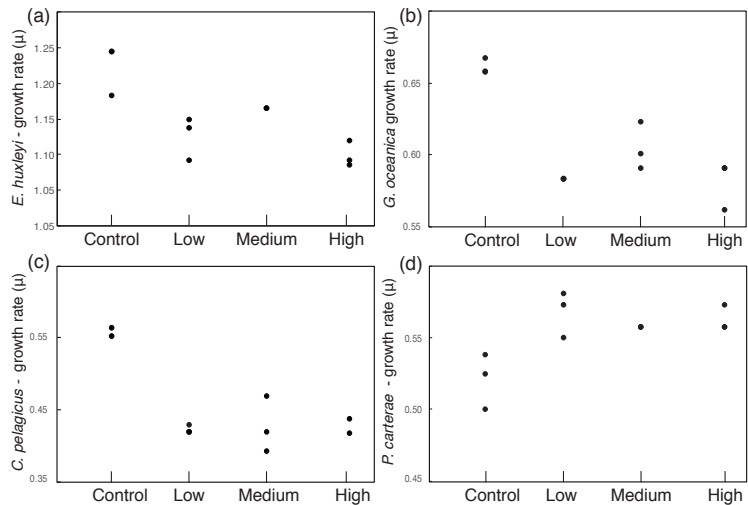

**Figure 4: Growth rate A)** *E. huxleyi;* **B)** *G. oceanica;* **C)** *C. pelagicus;* **D)** *P. carterae.* **Note the different scales on the y-axis.**





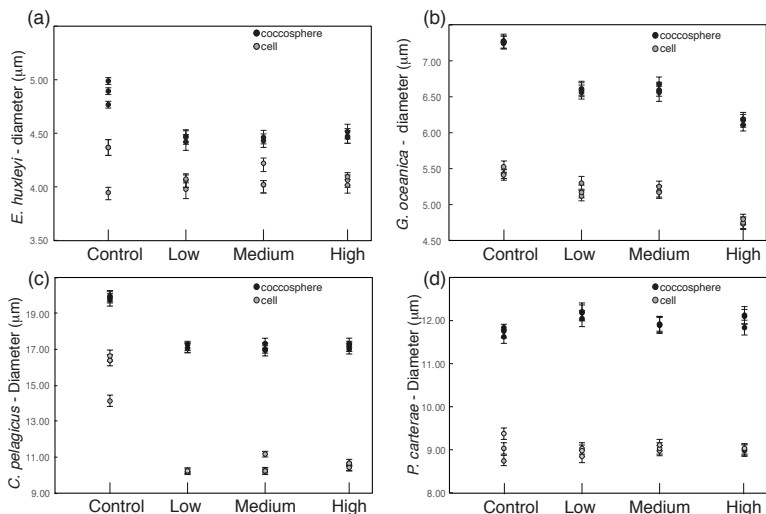

5  **Figure 5: Coccolith-bearing cell and coccolith-free cell diameters A)** *E. huxleyi;* **B)** *G. oceanica*; **C)** *C. pelagicus*; **D)** *P. carterae.* **Note the different scales on the y-axis.**
.

25





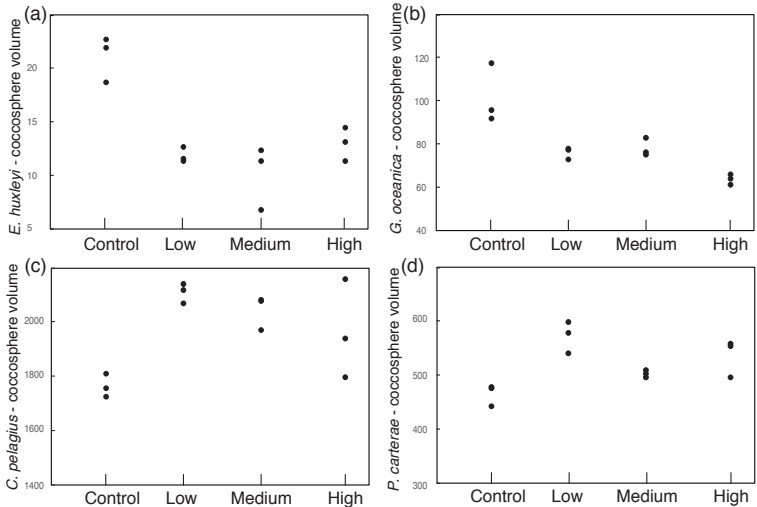

**Figure 6: Coccosphere volume: A)** *E. huxleyi;* **B)** *G. oceanica*; **C)** *C. pelagicus*; **D)** *P. carterae.* **Note the different scales on the y-axis.**




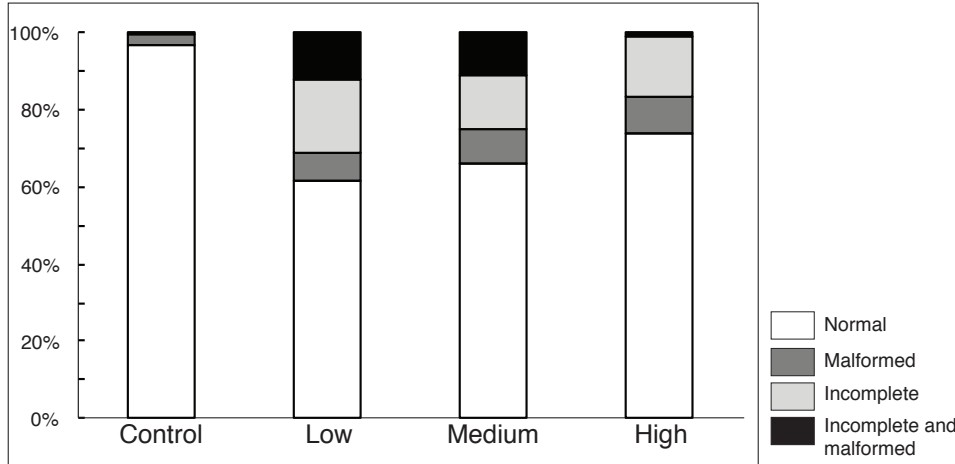

**Figure 7: Malformation percentage. Percentages of normal, malformed, incomplete and malformed and incomplete coccoliths of *E. huxleyi* versus trace metal concentrations.**

25



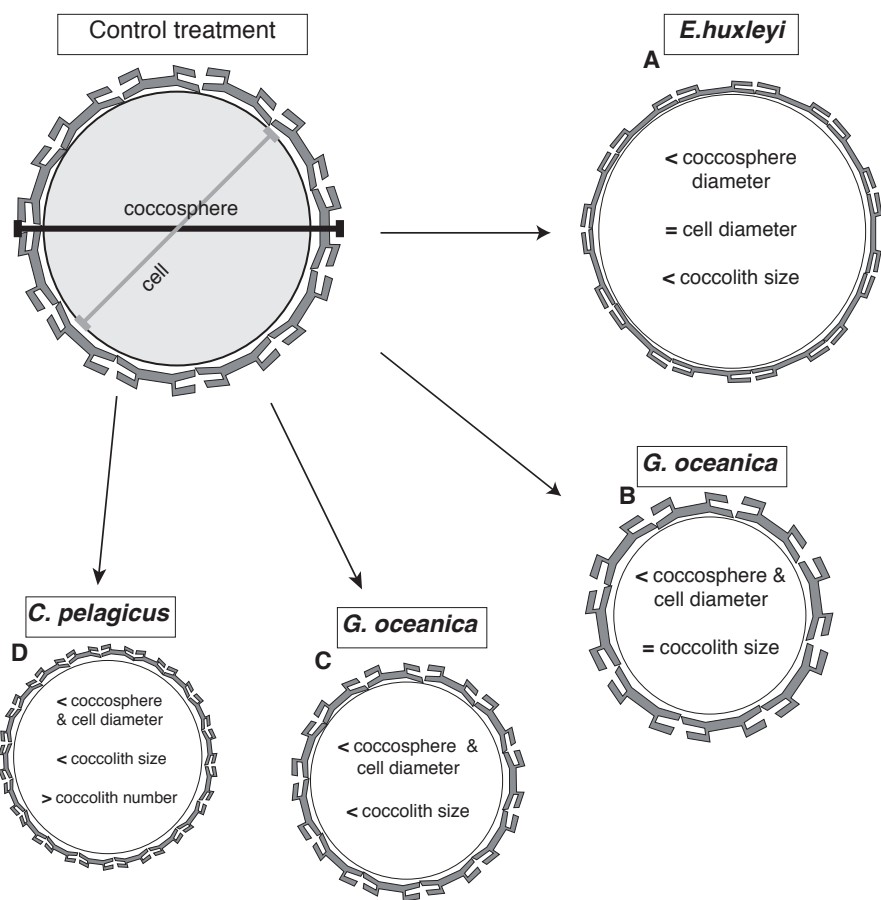

**Figure 8: Species-specific coccolith-bearing cell, coccolith-free cell and coccolith sizes response to trace metal enrichment**



| | Control | Low | Medium | High | Extreme |
|---|---|---|---|---|---|
| | $\mu$mol $^{L-1}$ | | | | |
| $FeCl_3 \cdot 6H_2O$ | 11.7 | 11.7 | 11.7 | 11.7 | 11.7 |
| $Na_2 \cdot 2H_2O$ | 11.7 | 11.7 | 11.7 | 11.7 | 11.7 |
| $CuSO_4 \cdot 5H_2O$ | 0.04 | 0.04 | 0.04 | 0.04 | 0.04 |
| $Na_2MoO4 \cdot 2H_2O$ | 0.03 | 0.03 | 0.03 | 0.03 | 0.03 |
| $CoCl_2 \cdot 6H_2O$ | 0.04 | 0.04 | 0.04 | 0.04 | 0.04 |
| $ZnSO_4 \cdot 7H_2O$ | 0.08 | 0.16 | 0.16 | 0.96 | 8.08 |
| Pb | ……… | 0.01 | 0.08 | 0.8 | 8.00 |
| $NiCl_2 \cdot 6H_2O$ | ……… | 0.08 | 0.08 | 0.8 | 8.00 |
| $VOSO_4$ | ……… | 0.08 | 0.08 | 0.8 | 8.00 |

**Table 1.** Trace metal concentrations in the growth medium of the different treatments.

25

30

35



| | E. huxleyi | | | | G. oceanica | | | |
|---|---|---|---|---|---|---|---|---|
| | Control | Low | Medium | High | Control | Low | Medium | High |
| μ | 1.22 | 1.12* | 1.16 | 1.10* | 0.66 | 0.58* | 0.60* | 0.58* |
| Coccosphere D | 4.88 | 4.45* | 4.44* | 4.48* | 7.25 | 6.58* | 6.60* | 6.14* |
| Cell D | 4.22 | 4.04 | 4.08 | 4.05 | 5.45 | 5.18* | 5.19* | 4.74* |
| coccosphere volume | 20.98 | 11.78* | 10.01* | 12.83* | 101.02 | 75.48* | 77.41* | 62.96* |
| | C. pelagicus | | | | P. carterae | | | |
| | Control | Low | Medium | High | Control | Low | Medium | High |
| μ | 0.56 | 0.42* | 0.43* | 0.43* | 0.52 | 0.57* | 0.56* | 0.57* |
| Coccosphere D | 19.82 | 17.12* | 17.05* | 16.85* | 11.70 | 12.11* | 11.88 | 11.99* |
| Cell D | 15.65 | 10.10* | 10.46* | 10.38* | 9.03 | 8.93 | 9.02 | 8.98 |
| coccosphere volume | 1760 | 2102* | 1859* | 1954* | 463 | 570* | 500 | 533* |

**Table 2.** Growth rate, coccosphere (μm), cell diameters (μm) and coccosphere calcification volume ($μm^3$) from the batch experiments. Significance was tested using an ANOVA and a Tukey post-hoc test ($p < 0.05$). Asterisks indicate significant difference from the control treatment.

25

30

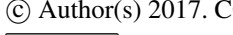



| | Control | | | Low | | | Medium | | | High | | |
|---|---|---|---|---|---|---|---|---|---|---|---|---|
| | DSL | DSW | DSA | DSL | DSW | DSA | DSL | DSW | DSA | DSL | DSW | DSA |
| *E. huxleyi* | 2.98 | 2.44 | 5.71 | 2.66* | 2.11* | 4.42 | 2.65* | 2.06* | 4.29 | 2.59* | 1.98* | 4.03 |
| *G. oceanica* | 4.31 | 3.77 | 12.78 | 4.10 | 3.57 | 11.47 | 4.15 | 3.58 | 11.69 | 3.97* | 3.37* | 10.49 |
| *C. pelagicus* | 12.76 | 11.08 | 111.08 | 10.43* | 8.43* | 69.11 | 10.03* | 8.17* | 64.38 | 10.47* | 8.50* | 69.87 |
| *P. carterae* | 1.90 | 1.18 | 1.75 | 1.68 | 1.05 | 1.38 | 1.87 | 1.16 | 1.70 | 1.90 | 1.19 | 1.78 |

5 **Table 3.** Coccolith DSL (µm) and DSW (µm) average values and calculated DSA (µm$^2$) for all experiments. Asterisks indicate significant difference from the control treatment. Significance of DSL and DSW was tested using an ANOVA and a Tukey post-hoc test ($p < 0.05$). Asterisks indicate significant difference from the control treatment

| | replicate | free coccolith number | | replicate | free coccolith number |
|---|---|---|---|---|---|
| **Control** | r1 | 2112 | **Medium** | r1 | 9017 |
| | r2 | 2297 | | r2 | 10046 |
| | r3 | 2972 | | r3 | 12325 |
| **Low** | r1 | 8876 | **High** | r1 | 13089 |
| | r2 | 7734 | | r2 | 11523 |
| | r3 | 8358 | | r3 | 11350 |

30

**Table 4.** Free *C. pelagicus* coccolith concentration measured with the coulter counter; r= replicates.