# Peer review of "Impact of trace metal concentrations on coccolithophore growth and morphology: laboratory simulations of Cretaceous stress."

_Biogeosciences, 2017_

## Referee Comment (RC1) · L.J. de Nooijer (Referee) · 23 May 2017

Dear editor,

After careful assessment of the manuscript BG-2017-138 by Faucher and co-workers on the effect of trace metals on coccolith growth, I recommend it for publication in Biogeosciences after moderate revisions. It is well-written and reports results that will be of interest to the audience of Biogeosciences. Below, I have listed my minor comments that I hope will help improving the manuscript.

There is one more serious issue I have with the content, which is the absence of data on the actual metal concentrations in the treatments. Why were those concentrations

not determined after preparing the culture media? It may have been that the concentrations did not vary much between treatments due to sorption of ions and this may therefore have important consequences for the interpretation of the data. I suggest that the authors either determine trace metal concentrations from the stock solutions, or explicitly report that the difference between treatments is inferred from the recipe that was used to make the culture media.

Sincerely,

Lennart de Nooijer

Abstract

Page 1, line 16: what does 'phylogenetically linked' mean?

Introduction

Page 2, line 4: are there references to support this statement?

Page 2, line 5: 'affect' here probably means 'negatively affect', consider replacing by e.g. 'hamper'. Is there a reference that has reported this?

Page 2, line 31: please add one/ a few references on the evolutionary relation between the studied species. Moreover, it is now suggested that the species themselves have separated from each other in the late Creataceous, whereas the extant species are likely much younger: the groups to which they belong may have separated in the late Cretaceous.

Page 3, line 6: no such studies

Page 3, line 9: metals

Methods

Page 3, line 30: what was the concentration of the EDTA in the trace metal stock solutions?

Page 4, line 3: what is meant by 'experimental conditions'? These are the conditions without the trace metals added, I assume.

Page 4, line 23: for how long were the samples incubated in 0.1 M HCl? Was this sufficient to dissolve all CaCO3?

Page 5, line 4: analysis

Page 5, line 17: comparison

Results

Page 6, line 5: although likely true, this is technically speaking an interpretation and should belong in the discussion.

Page 6, line 20: I don't understand the definition of coccosphere volume. Isn't the coccosphere simply the cell volume + the coccolith volume?

Page 6, line 26: 'cells' should be 'cell'

Page 7, line 9: 'the cells' should probably be 'cell volumes'

Discussion

Page 9, line 15: should be 'trace metals'

Page 9, line 17: should be hand

Figures

Please add to the caption what the individual dots and error bars represent.

---

## Referee Comment (RC2) · Anonymous Referee #2 · 31 May 2017

General comments

In this paper, Faucher et al. investigate the effect of various trace metal concentrations on the growth and morphology of four different coccolithophore species. Using laboratory experiments, the authors simulate the environmental stress identified in Mesozoic geological records and use four coccolithophore species phylogenetically related to Mesozoic species, for comparison with the fossil record. Based on the results obtained, the authors emphasize that each coccolithophore species responds differently to metal availability and that such species-specific response should be taken in to account when coccolithophore morphological characteristics are used to reconstruct seawater chemistry in the geological past. I read the review posted by Dr. L. J. de Nooijer and I agree

with his assessment. The manuscript is well written and the results presented can be of interest to a broad audience. However, there are some changes that I recommend the authors to consider in order to improve their manuscript. Overall, I recommend this article for publication in "Biogeosciences" after a minor to moderate revision.

Specific comments:

I agree with the comments already provided by Dr. L. J. de Nooijer. Below are few additional suggestions.

Abstract.

Page 1, line 20. The authors do not really discuss the changes in coccolithophore algae production as consequence of elevated trace metal concentrations in their experiments. Please, delete.

Introduction.

Page 2, line 19. "During the latest Cenomanian OAE 2 (. . ..), increased by about 8-20 times the background level". Is this "seawater background level"? Please, specify.

Page 2, line 24. Which coccolithophore species?

Page 2, lines 26-28. It is likely that this paper will be read by scientists, who might not be familiar with morphological phylogeny. I recommend adding few sentences to explain what morphological phylogeny is, its implications, and its relevance in this study.

Page 3, line 4. It might be worth to explain why E. huxleyi is so widely studied compared to other coccolithophore species.

Page 3, line 9. The trace metals tested – which ones?

Material and Methods.

Page 4, line 8. Please, provide the range of duration of each experimental treatment.

Page 4, line 10. What is meant by "main experiment"?

Page 5, section 2.4.2. Please, specify why these analyses were done only on E. huxleyi.

Results.

Page 6, lines 7-8. "On the other hand, E. huxleyi, G. oceanica, C. pelagicus and P. carterae survived in L, M and H". I suggest adding the word "treatments" (or equivalent) at the end of the sentence.

Page 6, lines 28-29. "The coccosphere volume was significantly reduced under increased trace metal concentrations compared to control conditions (Fig 6b) with similar coccosphere volumes recorded in both L, M and H". Can the coccosphere volume be really defined "similar" in L, M, and H?

Discussion.

As a general comment, there is no discussion of E. huxleyi coccolith malformations (Figure 3).

Figure 1. The meaning of the grey line is a little bit foggy – what does it represent? Does it represent the coccosphere volume? Please, specify.

Figure 2. I would recommend moving the column "Control" prior the columns "Low", "Medium", and "High", for consistency with Tables 1-4 and the other Figures.

Table 2. Growth rate, coccosphere diameter, cell diameter, and coccosphere volume are reported either as (almost) fully spelled name or as symbol. Please, be consistent and revise the table caption accordingly.

---

## Author Comment (AC1) · 21 Jun 2017

We greatly appreciate the valuable comments and critical reading of the manuscript made by L.J. de Nooijer and a second anonymous reviewer which were useful to improve the scientific quality of the manuscript. Please find below our answers to the Reviewers comments.

Dear editor,

After careful assessment of the manuscript BG-2017-138 by Faucher and co-workers on the effect of trace metals on coccolith growth, I recommend it for publication in Biogeosciences after moderate revisions. It is well-written and reports results that will be of interest to the audience of Biogeosciences. Below, I have listed my minor comments that I hope will help improving the manuscript. There is one more serious issue I have with the content, which is the absence of data on the actual metal concentrations in the treatments. Why were those concentrations not determined after preparing the culture media? It may have been that the concentrations did not vary much between treatments due to sorption of ions and this may therefore have important consequences for the interpretation of the data. I suggest that the authors either determine trace metal concentrations from the stock solutions, or explicitly report that the difference between treatments is inferred from the recipe that was used to make the culture media.

Authors reply: We have not analysed the metal concentration in our media. The difference between treatments was insured by adding adequate amounts of EDTA to the culture media to avoid precipitation of metal ions. It is, therefore, safe to consider that the concentrations of metals added to the culture media represent their dissolved ion concentrations. The ratio of EDTA to metals in our "high" treatment, for example, was well within the range of EDTA to the metal ratio used in other studies which test the effect of V (partly in combination with Mo and Fe) concentrations on phytoplankton species (Bellenger et al. 2008a, Bellenger et al. 2008b).

Abstract

Reviewers' comment: Page 1, line 16: what does 'phylogenetically linked' mean?

Authors reply: We improved the text in order to make it clearer. The phylogenetic history of coccolithophores shows that the selected/investigated living species are linked to Mesozoic species showing dwarfism under excess metal concentrations.

Introduction

Reviewers' comment: Page 2, line 4: are there references to support this statement?

Authors reply: References added

Reviewers' comment: Page 2, line 5: 'affect' here probably means 'negatively affect', consider replacing by e.g. 'hamper'. Is there a reference that has reported this?

Authors reply: References added and text modified accordingly Reviewers' comment: Page 2, line 31: please add one/ a few references on the evolutionary relation between the studied species. Moreover, it is now suggested that the species themselves have separated from each other in the late Cretaceous, whereas the extant species are likely much younger: the groups to which they belong may have separated in the late Cretaceous. Authors reply: Text revised and references added.

Reviewers' comment: Page 3, line 6: no such studies

Authors reply: Text modified accordingly

Reviewers' comment: Page 3, line 9: metals Authors reply: Text modified accordingly

Methods Reviewers' comment: Page 3, line 30: what was the concentration of the EDTA in the trace metal stock solutions?

Authors reply: The EDTA concentration in our culture media was 11.71 $\mu$M. We added this information to table 1.

Reviewers' comment: Page 4, line 3: what is meant by 'experimental conditions'? These are the conditions without the trace metals added, I assume.

Authors reply: Cultures were pre-exposed to the four-experimental conditions (normal, low, medium, high and extreme), considering an acclimation period of some generations. The text was modified accordingly.

Reviewers' comment: Page 4, line 23: for how long were the samples incubated in 0.1 M HCl? Was this sufficient to dissolve all CaCO3?

Authors reply: The samples of the present study were acidified and directly measured (within minutes). The coulter counter measurements evidence the disappearance of all free coccoliths (Fig. 1) after the treatment with acid. Furthermore, samples were

analyzed with a cross-polarizing microscope and, after the treatment with HCl, no coccoliths were left.

Reviewers' comment: Page 5, line 4: analysis

Authors reply: We made this change

Reviewers' comment: Page 5, line 17: comparison

Authors reply: We made this change

Results Reviewers' comment: Page 6, line 5: although likely true, this is technically speaking an interpretation and should belong in the discussion.

Authors reply: The text was revised and modified following the Reviewer suggestion.

Reviewers' comment: Page 6, line 20: I don't understand the definition of coccosphere volume. Isn't the coccosphere simply the cell volume + the coccolith volume?

Authors reply: We change the name "coccosphere volume" as "volume of the calcitic portion of the coccosphere" (VCP) (see page 4). The volume of the calcitic portion of the coccosphere" (VCP) was estimated as:

Volume of the calcitic portion of the coccosphere (VCP) = coccosphere volume - cell volume The coccosphere volume is the coccolith-bearing cell volume, while the cell volume is the coccolith-free cell volume.

Reviewers' comment: Page 6, line 26: 'cells' should be 'cell'

Authors reply: We made this change

Reviewers' comment: Page 7, line 9: 'the cells' should probably be 'cell volumes'

Authors reply: We made this change

Discussion Reviewers' comment: Page 9, line 15: should be 'trace metals'

Authors reply: We made this change

Reviewers' comment: Page 9, line 17: should be hand

Authors reply: We made this change

Figures Reviewers' comment: Please add to the caption what the individual dots and error bars represent.

Authors reply: Information added in the captions

Bellenger, J. P., T. Wichard, and A. M. L. Kraepiel. 2008a. Vanadium Requirements and Uptake Kinetics in the Dinitrogen-Fixing Bacterium Azotobacter vinelandii Applied and Environmental Microbiology 74:1478-1484. Bellenger, J. P., T. Wichard, A. B. Kustka, and A. M. L. Kraepiel. 2008b. Uptake of molybdenum and vanadium by a nitrogen-fixing soil bacterium using siderophores. Nature Geoscience 1:243-246.

Please also note the supplement to this comment:
http://www.biogeosciences-discuss.net/bg-2017-138/bg-2017-138-AC1-supplement.pdf

---

## Author Comment (AC2) · 21 Jun 2017

We greatly appreciate the valuable comments and critical reading of the manuscript made by L.J. de Nooijer and a second anonymous reviewer which were useful to improve the scientific quality of the manuscript. Please find below our answers to the Reviewers comments.

General comments: In this paper, Faucher et al. investigate the effect of various trace metal concentrations on the growth and morphology of four different coccolithophore species. Using laboratory experiments, the authors simulate the environmental stress identified in Mesozoic geological records and use four coccolithophore species phylogenetically related to Mesozoic species, for comparison with the fossil record. Based on the results obtained, the authors emphasize that each coccolithophore species responds differently to metal availability and that such species-specific response should be taken in to account when coccolithophore morphological characteristics are used to reconstruct seawater chemistry in the geological past. I read the review posted by Dr. L. J. de Nooijer and I agree with his assessment. The manuscript is well written and the results presented can be of interest to a broad audience. However, there are some changes that I recommend the authors to consider in order to improve their manuscript. Overall, I recommend this article for publication in "Biogeosciences" after a minor to moderate revision.

Specific comments:

I agree with the comments already provided by Dr. L. J. de Nooijer. Below are few additional suggestions.

Abstract.

Reviewers' comment: Page 1, line 20. The authors do not really discuss the changes in coccolithophore algae production as consequence of elevated trace metal concentrations in their experiments. Please, delete.

Authors reply: We made this change

Introduction.

Reviewers' comment: Page 2, line 19. "During the latest Cenomanian OAE 2 (. . ..), increased by about 8-20 times the background level". Is this "seawater background level"? Please, specify.

Authors reply: Yes, the text was modified accordingly.

Reviewers' comment: Page 2, line 24. Which coccolithophore species?

Authors reply: We added the nannofossil species. Although, size changes during Cretaceous OAEs are further examined in the discussion.

Reviewers' comment: Page 2, lines 26-28. It is likely that this paper will be read by scientists, who might not be familiar with morphological phylogeny. I recommend adding few sentences to explain what morphological phylogeny is, its implications, and its relevance in this study.

Authors reply: We have improved/modified the text following the suggestions of the Reviewer.

Reviewers' comment: Page 3, line 4. It might be worth to explain why E. huxleyi is so widely studied compared to other coccolithophore species.

Authors reply: We have improved the text following the suggestions of the Reviewer.

Reviewers' comment: Page 3, line 9. The trace metals tested – which ones?

Authors reply: The four trace metals tested (Ni, Zn, V and Pb) are listed in the material and method paragraph.

Material and Methods.

Reviewers' comment Page 4, line 8. Please, provide the range of duration of each experimental treatment.

Authors reply: Text modified accordingly.

Reviewers' comment Page 4, line 10. What is meant by "main experiment"?

Authors reply: We delated "main" in the text.

Reviewers' comment: Page 5, section 2.4.2. Please, specify why these analyses were done only on E. huxleyi.

Authors reply: We added a comment in the text.

Results.

Reviewers' comment: Page 6, lines 7-8. "On the other hand, E. huxleyi, G. ocean-ica, C. pelagicus and P. carterae survived in L, M and H". I suggest adding the word "treatments" (or equivalent) at the end of the sentence.

Authors reply: Text modified accordingly.

Reviewers' comment: Page 6, lines 28-29. "The coccosphere volume was significantly reduced under increased trace metal concentrations compared to control conditions (Fig 6b) with similar coccosphere volumes recorded in both L, M and H". Can the coccosphere volume be really defined "similar" in L, M, and H?

Authors reply: We checked the results and we modified the text accordingly.

Discussion.

Reviewers' comment: As a general comment, there is no discussion of E. huxleyi coc-colith malformations (Figure 3).

Authors reply: The Reviewer is correct. We improved the text following the suggestions of the Reviewer (page 7).

Reviewers' comment: Figure 1. The meaning of the grey line is a little bit foggy – what does it represent? Does it represent the coccosphere volume? Please, specify.

Authors reply: The grey line represents coccolith-free cell volume of C. pelagicus after acidification with HCl. The caption was modified.

Reviewers' comment: Figure 2. I would recommend moving the column "Control" prior the columns "Low", "Medium", and "High", for consistency with Tables 1-4 and the other Figures.

Authors reply: The figure was modified accordingly.

Reviewers' comment: Table 2. Growth rate, coccosphere diameter, cell diameter, and coccosphere volume are reported either as (almost) fully spelled name or as symbol.

Please, be consistent and revise the table caption accordingly

Authors reply: Table and caption modified.

Please also note the supplement to this comment:
http://www.biogeosciences-discuss.net/bg-2017-138/bg-2017-138-AC2-supplement.pdf

---

## Author Response (AR2)

**Dear Editor,**

we thank you for your critical reading of the manuscript "*Impact of trace metal concentrations on coccolithophore growth and morphology: laboratory simulations of Cretaceous stress*". We are really pleased with your final decision. We improved the manuscript with the technical corrections that you suggested.

With best Regards Giulia Faucher

[revised manuscript text omitted]